

# A cross-sectional study for the mental health status and sleep quality among college students in Macao during the COVID-19 pandemic

Xiaoyu Tao[1], Dong Chen[1], Yawen Fan[1], Lanxin Zhang[1], Houqian Shan[1], Yi Wei[2], Xi Yu[1], Tian Zhong[1], Ling Wang[1], Sookja Kim Chung[1], Yaqin Yu[3] and Ying Xiao[1]

[1] Faculty of Medicine, Macau University of Science and Technology, Macao, China
[2] School of Public Administration, Jilin University, Changchun, Jilin, China
[3] Faculty of Health Sciences, Zhuhai College of Science and Technology, Zhuhai, Guangdong, China

Corresponding authors
Ling Wang, lingwang@must.edu.mo
Ying Xiao, yxiao@must.edu.mo

## ABSTRACT

**Objectives:** The main focus of this study was to investigate the effect of the coronavirus disease (COVID-19) pandemic on the mental health condition and sleep quality of college students in Macao. In addition, the students' behaviours during the pandemic, such as drinking alcohol, taking sleeping pills, and seeking psychological counselling were analyzed.

**Method:** A cross-sectional survey of mental health and sleep quality status, as well as the possible behavioral risk factors, was conducted among the college students of Macao in August, 2020 during the COVID-19 pandemic. An online self-report questionnaire survey method was applied to assess the general demographics and related lifestyle behaviors of students. The general mental health condition and sleep quality were evaluated through the General Health Questionnaire (GHQ-12) and Pittsburgh Sleep Quality Index (PSQI) questionnaires, respectively. The main statistical methods included the Chi-square test, Bonferroni correction, and Pearson correlation. Data analysis was performed using SPSS Version 24.0.

**Results:** A total of 980 students were investigated in the study, of which 977 completed the survey. During the COVID-19 pandemic period, overall college students in Macao were psychologically well adjusted and reported good quality of sleep. However, female students were in poorer psychological condition than males ($P < 0.05$). Moreover, the students over 20 years of age had poorer sleep quality than students aged less than or equal to 20 ($P < 0.05$). The significant differences were found among the students in different study majors for the mental health status and sleep quality (both $P < 0.05$), which were associated with certain behaviors, such as drinking alcohol, taking sleeping pills, and seeking for help in psychological counselling during the COVID-19 pandemic period.

**Conclusions:** Poor mental health status could be either the consequence or cause of sleep disturbance, which might further affected physical health. Therefore, regular assessment of mental health condition and sleep quality of college students is

particularly necessary during public health emergencies, such as the COVID-19 pandemic, and appropriate intervention should be provided to the students.

# INTRODUCTION

The *World Health Organization (2020)* declared an outbreak of COVID-19 on 30th January 2020, which constituted a Public Health Emergency of International Concern (PHEIC) (*World Health Organization, 2020*). The severe acute respiratory syndrome coronavirus 2 (SARS-CoV-2) caused respiratory, cardiovascular and intestinal infections. It is highly contagious and could be transmitted through respiratory droplets and close contacts. The patients' clinical manifestations included asymptomatic symptoms, fever, dry cough, shortness of breath, and even respiratory failure leading to death (*Du Toit, 2020*; *Wiersinga et al., 2020*; *Zhong et al., 2020*).

The Chinese government implemented prevention and policies to control the spread of the epidemic. In China, it was mandatory for all citizens to wear masks and keep a proper social distance (*Horton, 2021*). Many regions were closed down, including Macao, which had banned people from entering and leaving the Macao special administrative regions (S.A.R.). For social restriction measures and limited exposure to pathogens, people were advised to work at home and study on-line.

Similar to other major public health emergencies, COVID-19 threatened people's physical health and caused a variety of psychological problems (*Lötsch et al., 2017*; *Qiu et al., 2020*; *Talevi et al., 2020*). During the COVID-19 pandemic, it was reported that there were more people with higher general mental health scores (GHQ-12) (*Ran et al., 2020*), which suggested that the mental health status of the public became worsen during the COVID-19 pandemic.

At the early stage of COVID-19 pandemic, it did not only affect the physical health, but also, led to emotional tension due to lacking the knowledge of dealing with the disease and urgent physical care, and increased psychological stress and mental illness in the public. Mental health services around the world took urgent measures to address these phenomena (*Moreno et al., 2020*). Although the community made the effort to provide appropriate psychological guidance to the public, people's mental health was inevitably affected during the pandemic due to the ineffective social communication. College students might be at higher risk of chronic mental health problems associated with an acute, high-intensity, and uncontrollable stress during the outbreak of the COVID-19, since they were under greater psychological stress resulting from lack of interpersonal communication, active face to face job hunting, online academic studies, and financial difficulties, (*Chi et al., 2020*). Such pressure may weaken people's sense of happiness and social well being, especially for college students, who showed negative emotions due to the suspension of classes (*Zhai & Du, 2020*). It was reported that people aged 18–30 and over 60 years old presented with the highest distress index during the COVID-19

peritraumatic condition (*Liang et al., 2020*). Disruptions of daily activities, such as group study, leisure, and family communication and social life, could also lead to boredom, depression, anxiety, fear, and other negative emotions among college students (*Khan et al., 2020*; *Liang et al., 2020*; *Thomas, 2021*). Sleep disorder was an important public health problem. During the COVID-19 pandemic, more people showed insomnia and other sleep disorders (*Huang & Zhu, 2020*; *Zhou et al., 2020*), which might lead to the poor psychological condition of people. Psychological distress among college students has been a concern for public health authorities around the globe. The unsatisfactory sleep also can lead to short and long-term negative impacts on the psychological well-being. It is necessary to investigate the mental health status and sleeping quality, as well as their relationship in college students of Macao during the COVID-19 pandemic and provide some guidelines for preventing mental and sleep disturbance associated with future public health emergencies. The current study analyzed the association among the mental health status, sleep quality, and the behaviors, including alcohol drinking and psychological help-seeking, in college students of Macao during the COVID-19 pandemic period through cross-sectional investigation.

## METHODOLOGY

### Subjects and procedures

The online survey was conducted among the college students of Macao in August 2020 before the commencement of the autumn semester. The participants were selected through snowball sampling. The questionnaire was distributed through the Wenjuanxing platform (Changsha Haoxing Information Technology Co., Ltd., Changsha, China). After the informed consent process, the participants completed the questionnaire on the mobile phone by scanning the QR (Quick Response) code *via* WeChat (a social network APP in China). The questionnaire was filled only once for each registered account holder and was not able to be modified after submission. The Ethical Review Committee of the Zhuhai M.U.S.T. Science Research Academy approved to carry out the study (Ethical Application Ref: MUST-MEC-20200701XY).

### Measurement

#### Demographic variables

The demographic variables included age (≤20 years and >20 years), gender (Male and Female), and study major (Humanities & Arts, Commercial, Law & Management, Science & Technology, Medicine & Health).

#### Mental health

The 12-item General Health Questionnaire (GHQ-12) is a well-validated, widely used, self-rated survey toll for detecting psychological morbidity and psychiatric disorders. The Chinese version of the GHQ-12 was used to detect the level of general mental health among the respondents. This questionnaire was developed by Goldberg and revised by scholar Taian Cheng (*Cheng & Williams, 1986*; *Goldberg & Williams, 1988*). The common method is binary GHQ scoring (0–0–1–1), which yields a possible score range of 0–12.

Higher scores indicate worse mental health status. GHQ-12 scores of four and above indicate a tendency towards the psychological disturbance.

### Status of sleep quality

Pittsburgh Sleep Quality Index (PSQI) (*Buysse et al., 1989*) was used to assess the sleep quality of participants. It consisted of 19 self-evaluation items and five other evaluation items. The 19 items constituted seven dimensions, and each component is on a scale from 0 to 3, with 0 indicating the highest sleep quality and 3 indicating the lowest one (*Rasekhi, Pour Ashouri & Pirouzan, 1970*; *Tsai et al., 2005*). The cumulative score from the individual dimensions is the total score of PSQI, which ranges from 0 to 21. The higher the overall score suggests, the worse sleep quality. The scores of 0–4, 5–7, and 8–21 indicate good, general, and poor sleep quality respectively.

### Life behaviors

The behavioral survey was a self-designed questionnaire based on literature review (*Ma & Lai, 2018*; *Pretorius et al., 2019*; *Shi et al., 2020*). It allowed to investigate the association between mental health status and respondents' behaviors, including smoking, alcohol consumption, the use of psychiatric medications, and seeking for psychological services on their own initiative.

## Data analysis

Data were analyzed with SPSS 24.0. Categorical data were analyzed by the chi-square test. Pearson correlation analysis using the demographic variables as independent variables and questionnaire scores as the outcome variable were conducted to identify the factors associated with mental health status. A two-tailed $P < 0.05$ was considered statistically significant.

## RESULTS

A total of 977 students completed the questionnaire. The proportion of aged over 20 years was less than the younger group. The ratio of males to females was 1.19. Details are presented in Table 1.

The mental health status was significantly different in various study majors ($P < 0.001$), and students majoring in Humanities & Arts had poorer mental health status than other majors ($P < 0.05$, Bonferroni corrected). More details are shown in Table 2. The score of GHQ-12 was significantly higher in female students than that of male ones ($P = 0.007$). On the other hand, 54.86% of participants had good sleep quality during the COVID-19 pandemic, suggesting that only 45.14% of participants had poor sleep quality.
The sleep quality had significant differences with age ($P = 0.004$) and majors ($P < 0.001$). Participants older than 20 had poor sleep quality. Moreover, students with the major in Medicine & Health had better sleep quality ($P < 0.05$, Bonferroni corrected). The results are shown in Table 3.

During the pandemic, the lifestyle behaviors, such as initiating alcohol drinking, taking sleeping pills, and seeking for help in psychological counselling, might affect mental health

**Table 1 Demographic characteristics of the participant ($n$ = 977).**

| Items | $n$ (%) |
|---|---|
| Age (year) | |
| ≤20 | 594 (60.80) |
| >20 | 383 (39.20) |
| Gender | |
| Male | 530 (54.25) |
| Female | 447 (45.75) |
| Major | |
| Humanities and Arts | 238 (24.36) |
| Commercial, Law and Management | 286 (29.27) |
| Science and Technology | 236 (24.16) |
| Medicine and Health | 217 (22.21) |

**Table 2 GHQ-12 scores of the participant ($n$ = 977).**

| | Good (%) | Poor (%) | $\chi^2$ | $P$ |
|---|---|---|---|---|
| Age | | | | |
| ≤20 | 310 (87.32) | 45 (12.68) | 0.393 | 0.531 |
| >20 | 552 (88.75) | 70 (11.25) | | |
| Gender | | | | |
| Male | 408 (91.28) | 39 (8.72) | 7.361 | 0.007 |
| Female | 454 (85.66) | 76 (14.34) | | |
| Major | | | | |
| Humanities & Arts | 195 (81.93) | 43 (18.07) | 20.338 | <0.001 |
| Commercial, Law & Management | 246 (86.01) | 40 (13.99) | | |
| Science & Technology | 221 (93.64) | 15 (6.36) | | |
| Medicine & Health | 200 (92.17) | 17 (7.83) | | |

**Table 3 PSQI scores of the participant ($n$ = 977).**

| | Good (%) | General (%) | Poor (%) | $\chi^2$ | $P$ |
|---|---|---|---|---|---|
| Age | | | | | |
| ≤20 | 350 (58.92) | 162 (27.27) | 82 (13.80) | 11.072 | 0.004 |
| >20 | 186 (48.56) | 122 (1.85) | 75 (19.58) | | |
| Gender | | | | | |
| Male | 263 (58.84) | 120 (26.85) | 64 (14.32) | 5.348 | 0.069 |
| Female | 273 (51.51) | 164 (30.94) | 93 (17.55) | | |
| Major | | | | | |
| Humanities & Arts | 109 (45.80) | 82 (34.45) | 47 (19.75) | 21.602 | 0.001 |
| Commercial, Law & Management | 155 (54.20) | 74 (25.87) | 57 (19.93) | | |
| Science & Technology | 134 (56.78) | 73 (30.93) | 29 (12.29) | | |
| Medicine & Health | 138 (63.59) | 55 (25.35) | 24 (11.06) | | |

**Table 4 Correlation between life behavior and general mental health status & sleep quality during the pandemic in the subjects.**

| | | GHQ-12 | PSQI | The behaviors during the COVID-19 pandemic | | | | | |
| --- | --- | --- | --- | --- | --- | --- | --- | --- | --- |
| | | | | The original cigarette consumption has increased | Try to start smoking | The original alcohol consumption has increased | Try to start drinking | Relieve neurological or psychiatric symptoms by pills | Seek psychological counselling |
| GHQ-12 | Pearson correlation coefficient | 1 | 0.259 | −0.274 | −0.032 | −0.008 | −0.034 | −0.169 | −0.318 |
| | P | | 0.073 | 0.057 | 0.327 | 0.936 | 0.315 | <0.001 | <0.001 |
| PSQI | Pearson correlation coefficient | 0.259 | 1 | 0.011 | −0.023 | −0.020 | −1.00 | −0.262 | −0.259 |
| | P | 0.073 | | 0.940 | 0.476 | 0.844 | 0.003 | <0.001 | <0.001 |

and sleep quality to a certain extent. The score of GHQ-12 and relieving neurological or psychiatric symptoms by pills and by psychological counselling were related ($P < 0.001$). Furthermore, students who used pills or sought counselling had higher GHQ-12 scores and poorer general mental health status than those who did not. The students with higher score of PSQI tended to initiate alcohol drinking and take the pills to relieve neurological or psychiatric symptoms ($P < 0.05$), and seek seeking psychological counselling ($P < 0.001$) and poorer sleep quality. The results are summarized in Table 4.

## DISCUSSION

In comparison to the other situation in Mainland China and other countries (*Jokela et al., 2020*; *Lin et al., 2021*; *Wu et al., 2020*), the results of this cross-sectional study showed that the general mental health status and sleep quality of college students in Macao were basically good during the COVID-19 pandemic. Only 11.77% of the participants were rated as having bad general mental health status. Such a positive result might be the consequence of the Macao S.A.R. government being actively concerned about the mental health of the populace and the general public during the pandemic. The Macao S.A.R. government created a good anti-pandemic environment by implementing good policies and imposing strict recommendations for the public based on rational scientific data. Generally, the public residents cooperated with the government to fight against the COVID-19 outbreak in Macao and the spread of this disease. At the same time, colleges in Macao attached importance to the psychological changes of students, ensured that college students could get psychological care and counselling in time when needed during the pandemic (*New Coronavirus Infection Strain Coordination Center, 2021*).

General mental health status was strongly correlated with gender during the COVID-19 pandemic in the investigated students ($P = 0.007$). Compared with males, females had poorer general mental health, which was consistent with previous research results (*Wu et al., 2020*). It could be seen that, females had more psychological stress and were more distressed during the pandemic; there was also a significant difference between

general mental health status and study major. The students who majored in Humanities & Art were under high psychological pressure. The first reason was that they might have less knowledge of the COVID-19 pandemic due to their professional background (*Peng et al., 2020*). The second possible reason was that the pandemic had increased people's social distance, and their profession required more interpersonal communication. Medical students had solid knowledge of the COVID-19 pandemic prevention, so their psychological pressure was lighter (*Zeng et al., 2019*).

The lack of space for physical activity, limited social interaction and psychological stress associated with fear of infection after the COVID-19 outbreak might reduce the sleep quality in college students and increase their negative emotion. The COVID-19 outbreak had an indirect impact on the mental health of young people, which could be mediated by sleep quality (*Zhang et al., 2020*). Unlike previous study (*Zhou et al., 2020*), there was no statistically significant gender difference in sleep quality in the college students of Macao. However, college students over the age of 20 showed a poorer sleep quality, possibly due to that the students in senior year face more pressure from making plans for higher degree program studies or employment issues (*Chi et al., 2020*). In addition, online courses might affect the quality of communication and interaction among students and their teachers. In some cases, the completion of the final year or higher degree thesis could not be completed as expected due to face to face execution of the project and discussion of research data. During the COVID-19 outbreak, many businesses closed down leading to higher unemployment. Graduates faced more competition for fewer positions when looking for jobs. Interestingly, Medical students' sleep quality was better than other major students, possibly because medical students are better informed about how to protect themselves during the pandemic, less worried and stressed (*Zeng et al., 2019*).

Students, who had poorer sleep quality and mental health issues initiated drinking alcohol, taking pills, and seeking internet counselling to relieve neurological or psychiatric symptoms. The unsupervised, self-initiated intervention against mental and sleep disorders of students can lead to more disastrous outcomes. To maintain the good mental health of college students, it was recommended that the colleges' health care unit should actively investigate the mental health status of students, especially for senior students, and encourage students to seek psychological counselling during the pandemic. According to a survey, students who had undergone psychological counselling during the epidemic were generally in a poor state of mental health (*Liang et al., 2020*). Students could also appropriately use the internet for consultation. However, young people had different evaluations of online psychological counselling services (*Pretorius et al., 2019*). It is recommended to enrich the resources of internet psychology and improve their qualities. In addition, mental health clinics of schools should identify students seeking alcohol drinking or seeking pills to relieve their mental state and investigate the students' mental problems as early as possible to provide face to face or online counselling. During the COVID-19 pandemic, students should be recommended to maintain healthy lifestyle with good living habits, which may improve mental state and sleep quality.

## CONCLUSIONS

Poor mental health could be either the consequence or cause of sleep disturbance, which might further affect physical health. Therefore, regular assessment of mental health status and sleep quality of college students is particularly necessary during public health emergencies, such as the COVID-19 pandemic, and appropriate intervention should be provided to the students.

## LIMITATIONS

The current study had several limitations. First, it was a cross-sectional survey, which could not explain the cause-and-effect relationship between the COVID-19 pandemic and mental health status or sleep quality. Then, the study population was not a random sample and might not avoid bias in subject selection. Lastly, the questionnaire was finished in the form of a self-report; therefore, some of the answers might be subjective.

## ACKNOWLEDGEMENTS

We would like to thank all the participants who participated in this study.

### Funding

This work was supported by the Higher Education Fund of the Macao SAR Government (Grant No. HSS-MUST-2020-6). The funders had no role in study design, data collection and analysis, decision to publish, or preparation of the manuscript.

### Grant Disclosures

The following grant information was disclosed by the authors:
Macao SAR Government: HSS-MUST-2020-6.

### Competing Interests

The authors declare that they have no competing interests.

### Author Contributions

- Xiaoyu Tao conceived and designed the experiments, performed the experiments, analyzed the data, prepared figures and/or tables, authored or reviewed drafts of the paper, and approved the final draft.
- Dong Chen conceived and designed the experiments, performed the experiments, prepared figures and/or tables, and approved the final draft.
- Yawen Fan performed the experiments, prepared figures and/or tables, and approved the final draft.
- Lanxin Zhang performed the experiments, prepared figures and/or tables, and approved the final draft.
- Houqian Shan performed the experiments, prepared figures and/or tables, and approved the final draft.
- Yi Wei analyzed the data, prepared figures and/or tables, and approved the final draft.

- Xi Yu analyzed the data, prepared figures and/or tables, authored or reviewed drafts of the paper, and approved the final draft.
- Tian Zhong analyzed the data, authored or reviewed drafts of the paper, and approved the final draft.
- Ling Wang performed the experiments, authored or reviewed drafts of the paper, and approved the final draft.
- Sookja Kim Chung conceived and designed the experiments, authored or reviewed drafts of the paper, and approved the final draft.
- Yaqin Yu conceived and designed the experiments, performed the experiments, authored or reviewed drafts of the paper, and approved the final draft.
- Ying Xiao conceived and designed the experiments, performed the experiments, authored or reviewed drafts of the paper, and approved the final draft.

## Human Ethics

The following information was supplied relating to ethical approvals (*i.e.*, approving body and any reference numbers):

The Ethical Review Committee of the Zhuhai M.U.S.T. Science Research Academy approved to carry out the study (Ethical Application Ref: MUST-MEC-20200701XY).

## Data Availability

The raw measurements are available in the Supplemental File.

## Supplemental Information

Supplemental information for this article can be found online at http://dx.doi.org/10.7717/peerj.12520#supplemental-information.

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
