# Peer review of "A cross-sectional study for the mental health status and sleep quality among college students in Macao during the COVID-19 pandemic"

_PeerJ, doi:10.7717/peerj.12520_

## Round 0.1 · original submission · Major Revisions

The paper needs substantial revisions.

Reviewer 1 ·

Basic reporting

1. First if all, English language of this paper is poor and needs substantial revisions. In particular, several sentences an terms are not used in a professional way. In the title, “mental status” should be “mental health status” but this term is inaccurate because GHQ-12 and PSQI are measures of mental health and sleep problems and positive mental health such as happiness are not assessed.
2. Abstract. In the objectives part, the current study can not answer the questions of the effects of pandemic and behavioral factors on the mental health of college students because this is a cross-sectional study. The conclusion is not appropriate because prelateship between sleep and mental health is not the focus of this study.
3. Introduction. This part should be rewritten because it talked a lot of irrelevant information. Please indicate the public health significance of this research topic, mental health and sleep quality of college students in Macau during the pandemic, review existing studies, and comments on their limitations to indicate the necessity of this research topic.
4. Methodology. There is no “Conditions of Mental Health” term in psychiatry and mental health. Please understand GHQ-12 by reading related literature and rewrite this part. Line 111-112, the three cot-off scores are not suitable for Chinese populations. Please check relevant studies. The lifestyle factors are poorly described and seeking psychological services should not be a lifestyle factor.
5. The data analysis should be re-done. I suggest the authors to use binary outcomes: the presence of common mental health problems and poor sleep quality. Second, analyzing factors associated with common mental health problems and poor sleep quality. Third, analyzing the psychological services utilization of college students with .mental health problems and poor sleep quality.
6. I decided not to review the remainding parts due to the above concerns.

Experimental design

See above

Validity of the findings

See above

Reviewer 2 ·

Basic reporting

1. The reference of Chinese version GHQ-12 should be cited in section 2.2.2.
2. If a reference is published in Chinese, the title of the article should be followed with “(in Chinese)” in the reference list.
3. In line 100, the GHQ-12 is used for screening general mental health problems, not for detecting psychological morbidity and psychiatric disorders.
4. In line 111, “the higher the overall score, the worse the sleep quality” should be “the higher the overall score is, the worse the sleep quality is”.
5. In lines 117-118, does “drug use” refer to “the use of psychiatric medications”? Drug use in English context usually refers to addictive, hallucinogenic and illegal drug.
6. In line 144, “relieved neurological or psychiatric symptoms” should be “relieving neurological or psychiatric symptoms”.
7. In line 147, “Tried to start drinking” should be “Trying to start drinking”.
8. In line 148, “relieve neurological or psychiatric symptoms” should be “relieving neurological or psychiatric symptoms”.
9. Tables 1-3 could be combined to one table.

Experimental design

1. Sleep quality was grouped into good, general and poor in lines 111-112. However, in lines 136-138, sleep quality was only grouped into good and poor. Please make it clear.
2. In Table 4, were age, gender and major entered into the models as covariates? If not, it is strongly suggested to do so.

Validity of the findings

1. Please indicate where continuous correction chi-square test was used in Results section and in Tables.
2. In line 132, what is the cutoff score of GHQ-12 for poor mental health condition? Please clarify in Methods section. The 95% confidence interval of the proportion (11.77%) should be indicated.
3. In line 134 and line 140, the chi-squares for multiple comparisons are lacking. It is strongly suggested that the authors create other tables to demonstrate the results of multiple comparisons, including chi-squares, p-values, and Bonferroni-corrected significance level.

---

## Round 0.2 · accepted · Accept

Thanks for the revisions.

Reviewer 1 ·

Basic reporting

The English language is clear.

Experimental design

The design is methdologically acceptable.

Validity of the findings

The findings support the main conclusion.

Additional comments

I am satisfied with the revisions.

Reviewer 2 ·

Basic reporting

The authors have made reasonable revisions. No further comments.

Experimental design

The authors have made reasonable revisions. No further comments.

Validity of the findings

The authors have made reasonable revisions. No further comments.